# Effect of Environmental and Intrapersonal Factors on Leisure-Time Physical Activity of Chinese Rural Older People in Zhejiang Province

**DOI:** 10.3390/healthcare13111244

**Published:** 2025-05-25

**Authors:** Jiabin Yu, Jiayi Zhou, Xiaoguang Zhao, Huiming Huang, Beibei Si, Dawei Zhang, Chuang Gao, Junqi Jia

**Affiliations:** 1Faculty of Sport Science, Research Academy of Grand Health, Ningbo University, Ningbo 315211, China; 2Department of Physical Education, Central South University, Changsha 410017, China; nbuzhoujiayi@outlook.com

**Keywords:** elderly, self-efficacy, built environment, village, public health

## Abstract

**Objective:** The purpose of this study was to investigate the association of built environment and intrapersonal factors with leisure-time physical activity (LTPA), and the possible moderating effect of intrapersonal factors on the association between the built environment and LTPA. **Methods:** A total of 514 older people in the village of Fuwen were included in this study. Data on LTPA, intrapersonal factors (social, fitness, medical, benefits/challenges, recognition), and the built environment (traffic safety, street connectivity, walking facilities, access to services, crime safety, aesthetics, residential density, land use mix diversity) were collected. Multivariate linear regression analysis with the “enter” method was conducted to analyze the association of perceived scores of built environment and intrapersonal factors with LTPA. **Results:** Among the intrapersonal factors, fitness was positively related to LTPA (*p* = 0.012). Among the built environment factors, traffic safety (*p* = 0.02) and crime safety (*p* = 0.047) were positively related to LTPA, while walking facilities was negatively associated with LTPA (*p* = 0.007). Additionally, fitness had a significant moderating effect on the association between walking facilities and LTPA (*p* = 0.025). **Conclusions:** Older people with higher fitness requirements and better perceived traffic safety and crime safety tend to engage in greater levels of LTPA. The findings provide valuable insights for policymakers, particularly in designing ecologically livable villages that promote physical activity among older populations.

## 1. Introduction

Population aging is a serious social problem facing China. According to the latest data published by the National Bureau of Statistics of the Chinese government, the percentage of older people accounted for 22.0% at the end of 2024, which indicates that China has entered the stage of “deep aging”. Furthermore, the situation of population aging is more serious in villages than it is in cities [1]. Population aging causes a series of social problems, such as the medical burden on the government. According to the report published by the National Health Insurance Agency of the Chinese government in July 2024, the proportion of retirees’ medical expenses in the total cost of employees’ health insurance was close to 50 percent in China in 2023 [2]. Therefore, maintaining and enhancing the health of older people is important to decrease the medical burden on the government, and this has been emphasized in official policies of the Chinese government, such as “The 14th Five-Year Plan For Healthy Aging” [3]. With the accelerated process of urbanization, a large number of rural young and middle-aged laborers are flowing into cities, leading to an increase in the proportion of the elderly population in rural areas. Additionally, compared to the diversity of urban old-age care, rural care relies primarily on family care. Therefore, maintaining the health of older people living in villages is even more crucial.

The health benefits of physical activity have sufficient scientific evidence [4,5,6,7], and are recognized by the World Health Organization (WHO) [8,9]. As recommended by WHO, the health outcomes of older adults aged over 65 years include improved physical and mental health, helping to prevent falls and declines in bone health and functional ability [8]. Therefore, encouraging older people to take part in exercise and increase their level of physical activity is vital for healthy aging. As previous studies suggested, the influencing factors of physical activity can be summarized into three categories, including intrapersonal factors [10,11,12,13,14], interpersonal factors [10,15,16,17,18,19,20], and environmental factors [21,22,23,24,25,26].

For the built environment, the influencing elements vary for people in different age groups. In a recently published narrative review, Zhang et al. analyzed 29 reviews with moderate to high quality and suggested that pedestrian-friendly features and general safety were positively associated with the physical activity of children and older people, the availability and accessibility of shops/commercial services and parks/open access were positively related to the physical activity of adults and older people, and the walkability index was positively associated with physical activity in every age group [22]. The influencing elements also vary in places with different socioeconomic statuses (SESs). In a review study published in 2024, Hoyer-Kruse et al. reviewed 72 studies and highlighted the significant role of socioeconomic factors and the quality of physical activity infrastructure in promoting or hindering physical activity [26]. For example, regarding walking infrastructure, higher-income areas are often perceived as more aesthetically pleasing, with fewer physical barriers to walking and lower levels of crime and traffic [27], but low-SES areas tend to have poorer built environment experiences [28]. These differences have an effect on the association between the built environment and physical activity. For example, Zang et al. suggested that the physical activity of people living in low-SES areas was more dependent on the built environment compared to people living in high-SES areas [29].

The influence of intrapersonal and interpersonal factors on physical activity has been demonstrated by previous studies. Spiteri et al. analyzed 55 articles and suggested that social influences, reinforcement, and assistance in managing change were the most commonly identified motivators of physical activity in older people [30]. In a randomized clinical trial, McMahon et al. found that older people receiving a physical activity intervention with interpersonal behavior change strategies exhibited greater increases in total physical activity levels than those who did not at 1 week, 6 months, and 12 months after the intervention [10]. In China, Wang et al. analyzed the collected data of 737 older people from the province of Sichuan and suggested that self-regulation and social capital could directly affect LTPA [31]. The moderating effects of intrapersonal and interpersonal factors were also reported in some previous studies. Yiming et al. investigated the role of health locus of control, social support, and self-efficacy in adolescents’ physical activity and suggested that both self-efficacy and social support could directly affect physical activity. Furthermore, locus of control can indirectly affect physical activity through self-efficacy and social support [16]. In a Chinese study, Ren et al. also focused on adolescents and found that social support and self-efficacy were positive predictors of physical activity, and self-efficacy was a significant mediator in the relationship between social support and physical activity [12].

Although the effect of built environment and intrapersonal factors on physical activity has been well demonstrated by numerous studies, the effect might vary in different age groups, areas with different SESs, and so on. Therefore, the best way to identify the influencing factors of residents’ physical activity is based on locally collected data. In China, researchers have focused more on the effect of the built environment on physical activity [21,23,32,33,34,35,36,37,38], and less on the effect of intrapersonal factors on physical activity [12,31,39]. The participants in Chinese studies predominantly live in cities [21,23,31,34,35,36,40], with less attention paid to residents living in villages [15]. Therefore, the purpose of this study was to investigate the effect of built environment and intrapersonal factors on the LTPA of older people living in a Chinese village. We also attempted to explore the possible moderating effect of intrapersonal effects on the association of the built environment with LTPA. Based on previous studies, the hypothesis of this study was that some elements of built environment and intrapersonal factors would have an effect on the LTPA level of rural older people, and intrapersonal factors would act as a moderator of the association relationship between LTPA and built environmental factors. The findings of this study provide suggestions for the construction of ecologically livable villages, which has been emphasized by the Chinese government in the “Comprehensive Rural Revitalization Plan (2024—2027)”.

## 2. Materials and Methods

### 2.1. Process of Data Collection

The raw data of this study were collected in the village of Fuwen, which is located in the city of Hangzhou, Zhejiang province. Zhejiang province is located in the southeast coastal region of China and is one of the most economically developed provinces in China. It is also a region with profound historical and cultural heritage as well as beautiful natural scenery. This village was included in the list of beautiful rural demonstration villages of Zhejiang province in 2022. Regarding recreational facilities, there is a sports culture square available for the residents to engage in activities such as square dancing and Tai Chi. There is also a fitness park including some physical exercise facilities like single and parallel bars, waist twisters, and cycling machines to meet the residents’ daily physical activity needs. Furthermore, a landscaped fitness trail spanning 2.3 km is available for residents to walk and jog. A cross-sectional study of random samples was conducted in May 2023 and was approved by the ethics committee of the Research Academy of Grand Health, Ningbo University (RAGH20230500012, May 2023). Two researchers conducted face-to-face interviews to collect the data. Both researchers were trained prior to the interviews to ensure that they fully understood the questionnaires, data collection process, and announcements. The village committee of Fuwen village assisted in the data collection by helping to recruit older people and organize interviews. The inclusion criteria for the participants of this study were as follows: (a) older people aged over 60 years old; (b) an official resident of Fuwen village and having lived there for at least one year; (c) no cognitive impairment and possessing the ability to communicate and understand. Data from 523 older people were collected in the interviews. Data from 514 older people were included in this study after excluding the data of 9 individuals with missing parts to the questions of the questionnaire.

### 2.2. Questionaries Used in Data Collection

The data collected in this study included demographic characteristics, LTPA, perceived scores of intrapersonal factors, and perceived scores of the built environment. For demographic characteristics, the variables included age, gender, self-reported health status, the situation of lower extremity motor system disease, body height (BH), and body weight (BW). There were three options for self-reported health status: excellent, good, and poor. The two options for the situation of lower extremity motor system disease were yes and no. The unit of BH was meters, and the unit of BW was kilograms. Body mass index (BMI) was computed with the data of BH and BW based on the equation BW/BH2. The International Physical Activity Questionnaire—Short Form (IPAQ-S) was used to collect raw data on LTPA. Because the focus of this study was LTPA, the questions about sitting behavior were not included in our interview. Three questions about LTPA were included to collect data on walking, moderate-intensity physical activity, and vigorous-intensity physical activity. The details of this questionnaire can be found in the Appendix A section.

The perceived scores of intrapersonal factors were collected using the Participation Motivation Questionnaire for Older Adults (PMQOA). This questionnaire was originally developed from the Participation Motivation Questionnaire (PMQ), an instrument to assess the motives of youth for taking part in sports. For groups of older people, physical activity researchers modified the items from PMQ to develop the PMQOA, and established its validity [41]. PMQOA uses a 3-point Likert scale (1—totally disagree; 2—partly agree; 3—totally agree), and covers five elements, including social, fitness, medical, challenges/benefits, and recognition. The details of this questionnaire can be found in the Appendix A section. The perceived scores of built environment factors were obtained using the Neighborhood Environment Walkability Scale for Chinese Seniors (NEWS-CS). The reliability and validity of this questionnaire have been confirmed by the study group of Eser Cerin [42]. The Neighborhood Environment Walkability Scale (NEWS) is the most frequently and internationally used instrument to assess perceived attributes of neighborhood environments by researchers all around the world [37,40,43,44]. NEWS-CS uses 5-point Likert scales, and participants choose 1 point from 1 to 5 according to the agreement degree (1—totally disagree; 5—totally agree). Eight elements are contained in the NEWS-CS, including traffic safety, street connectivity, walking facilities, access to services, crime safety, aesthetics, residential density, and land use mix diversity. The details of this questionnaire can be found in the Appendix A section.

### 2.3. Statistical Analysis

Descriptive statistical analyses were performed to show the mean, standard deviation, and 95% confidence interval values for the demographic characteristics, LTPA, the perceived scores of intrapersonal factors, and the perceived scores of the built environment. Frequency analysis was also used to describe the demographic characteristics. Multivariate linear regression analysis was conducted to examine the association of perceived scores of built environment and intrapersonal factors with the LTPA of older people. The “enter” regression method was used. Before conducting the regression analysis, we conducted correlation analysis and confirmed that eight elements of the built environment and five elements of intrapersonal factors were significantly related to the LTPA level of older people. In the regression model of LTPA with built environment, the dependent variable was LTPA, and the independent variables were the eight elements of the built environment. In the regression model of LTPA with intrapersonal factors, the dependent variable was LTPA, and the independent variables were the five elements of intrapersonal factors. Multivariate linear regression analysis was also conducted to identify the possible moderating effect of intrapersonal factors on the association between LTPA and the built environment. In this regression model, the dependent variable was LTPA, and the independent variables were the eight elements of the built environment and the intrapersonal factors that were significantly related to LTPA. Statistical significance was set at *p* < 0.05. All the analyses were carried out using SPSS 19.0 software (IBM Inc., Chicago, IL, USA).

## 3. Results

Table 1 shows the demographic information of the older people included in this study. The average age was 67.5 years, and 66.9% of the participants were in the range of 60–69 years old. The percentage of men and women was almost equal. The health status was good for most of the older participants. A total of 70% of older people reported being free from a lower extremity motor system disease.

Table 2 presents the LTPA level of older people in the village of Fuwen. The level of leisure-time walking was higher than that of leisure-time moderate and vigorous physical activity.

Table 3 shows the self-reported scores of intrapersonal factors. The scores of the five elements were quite similar, and the fitness and medical elements showed slightly higher scores.

Table 4 shows the perceived scores of built environment factors. The score for crime safety was the highest, and the score for access to services was the lowest.

Table 5 displays the association results of intrapersonal factors with LTPA. The fitness element was positively related to LTPA at a significant level. The other four elements were not significantly related to LTPA.

Table 6 shows the association results of built environment elements with LTPA. Three built environment elements were significantly associated with LTPA. Traffic safety and crime safety were positively associated with LTPA, and walking facilities was negatively associated with LTPA. Other factors were not found to be related to LTPA.

Table 7 presents the association results of the built environment with LTPA with the moderating effect of an intrapersonal factor. As an intrapersonal factor significantly related to LTPA, fitness was the moderator of the relationship between LTPA and walking facilities. Fitness strengthened the negative association of walking facilities with LTPA.

## 4. Discussion

The purpose of this study was to investigate the effect of the perceived built environment on the LTPA of older people in the Chinese village of Fuwen, and also to explore the effect of intrapersonal factors on the LTPA of older people. We also aimed to determine whether intrapersonal factors act as moderators that influence the association of the built environment with the LTPA of older people.

### 4.1. The LTPA Level of Older People in This Village

We found that the average LTPA level of older people in the village of Fuwen was 1347.9 MET minutes per week (Table 2). This physical activity level is substantially lower than the level of older people living in cities [32,34,35,36]. Yu et al. reported that the LTPA levels of older people living in the cities of Hangzhou and Wenzhou were 2048.1 and 2676.7 MET minutes per week, respectively [40]. The lower LTPA of older people living in this village might be due to the fewer parks and other recreational facilities available. In a systematic review study, Van Cauwenberg et al. found evidence of positive relationships between LTPA and access to recreational facilities (*p* = 0.01) and parks/open space (*p* = 0.04) [45]. Leisure-time walking was the predominant form of LTPA among older people in this study. The average leisure-time walking level was 814.0 MET minutes per week, but the average leisure-time moderate and vigorous physical activity level was only 533.9 MET minutes per week. The World Health Organization recommends that older adults should engage in 150–300 min of moderate-intensity aerobic physical activity or 75–150 min vigorous-intensity aerobic physical activity, considering the health benefits of moderate and vigorous physical activity [8]. Clearly, older people in the village of Fuwen did not meet the recommended standard of moderate and vigorous physical activity. Therefore, identifying the factors of older people’s LTPA is crucial to increasing their LTPA levels.

### 4.2. Effect of Built Environment on LTPA of Older People

For the built environment, the results indicated that traffic safety, walking facilities, and crime safety were significant factors influencing the LTPA of older people in the village of Fuwen. Traffic safety was a positive factor of LTPA (Table 6), suggesting that the better perceived traffic safety was associated with higher levels of LTPA among older people. This finding is consistent with previous studies [21,25,46]. In a recently published review study [25], Mueller et al. suggested that for adults living in rural areas, sidewalks or streets with a low traffic volume acted as environmental facilitators for exercise and reaction [47], while a lack of sidewalks, speeding traffic, and high traffic volumes were barriers to exercise [48]. In the Chinese village of Fuwen, we observed that the roads are narrower than those in urban areas, and that sidewalks are not available for residents. In addition, the number of private cars is increasing rapidly due to economic development and government policies that encourage automobile consumption. These factors collectively decrease traffic safety for older people living in Fuwen, and hinder their participation in LTPA. Therefore, for local policymakers of Fuwen, improving the traffic safety, such as constructing sidewalks, widening roads, and separating pedestrians from vehicles, might encourage older people to engage in more LTPA.

Crime safety was another significant factor of the LTPA of older people in the village of Fuwen, and it was positively associated with the LTPA of older people. The result indicated that the better the perceived crime safety, the more LTPA older people will engage in. This finding is in line with previous studies [40,43]. In an umbrella review study, Bonaccorsi et al. analyzed 11 previous review studies and suggested that crime-related safety was an element positively associated with the physical activity of older adults [43]. A similar finding was also found for older people living in a city. Cerin et al. interviewed 484 older people living in Hong Kong and found that the odds of non-participant in LTPA were related to the safety aspects of the neighborhood [49]. This result is in conflict with a study conducted in the Chinese city of Xi’an. Sun et al. found a negative association relationship between adults’ LTPA and crime safety [36]. A possible explanation for the conflicting results might be the difference in participants. The average age of participants was 46.9 years old in Sun’s study, but it was 67.5 years old in our study. Older people might be more concerned about crime safety due to their lower flexibility and mobility. For local policymakers of Fuwen, efforts should be made to improve crime safety during both the daytime and nighttime. This would help older people feel safer when participating in LTPA outdoors. Implementing such policies could motivate older people to engage in more LTPA.

Walking facilities was a negative factor of the LTPA of older people in the village of Fuwen, and this means that the better the perceived walking facilities, the less LTPA older people will engage in. This result is surprising and in conflict with previous studies [25,50]. Frost et al. suggested that sidewalks are an environmental facilitator of physical activity for adults [51]. Similarly, Yun et al. suggested that walking amenities are a positively related environmental factor for the walking behaviors of older adults [50]. However, in a recently published review study, Hoyer-Kruse et al. suggested that when considering the effect of walking infrastructure on physical activity, the socioeconomic status (SES) of the surveyed areas should be taken into account. Compared with high-SES areas, low-SES areas tend to have poorer walking infrastructure, lower walkability scores, and less favorable structural attributes for physical activity. Furthermore, Hoyer-Kruse et al. emphasized that three studies found that living in high-SES areas was closely related to higher physical activity levels, but no association between the walk score and physical activity for those living in low-SES areas was found [26]. In our opinion, a possible explanation for the negative association might be that older people who took part in more LTPA had more opportunities to experience the walking facilities than older people engaged in less LTPA in the village of Fuwen, and they had a higher possibility of giving a lower perceived score of walking facilities in the interview. Nevertheless, walking facilities should be improved to provide a better environment for older people participating in LTPA in the village, as previous studies have suggested [46]. Whether the association between walking facilities and LTPA would be positive in other Chinese villages with better walking facilities needs to be confirmed in future studies.

### 4.3. Effect of Intrapersonal Factors on LTPA of Older People

For intrapersonal factors, fitness was a significant factor influencing the LTPA of older people in the village of Fuwen, while other intrapersonal factors were not significant. Fitness was positively associated with the LTPA of older people (Table 5), indicating that the higher the fitness requirements of older people, the more LTPA older people will engage in. This finding is consistent with previous studies [10,52,53]. McMahon et al. suggested that older people who received intrapersonal behavior change strategies exhibited greater increases in total physical activity than those who did not at 1 week, 6 months, and 12 months after the interventions [10]. In a systematic review, Macniven et al. suggested that improving one’s physical condition was the main motivator for the physical activity of older people [54]. Yarmohammadi et al. mentioned that important motivators for physical activity were more closely related to intrapersonal factors than to the interpersonal and environmental domains [53]. Therefore, for local policymakers of Fuwen, enhancing the publicity of the health benefits of physical activity and increasing intrapersonal motivation are vital to improving the LTPA level of older people.

### 4.4. Moderating Effect of Intrapersonal Factors

For the moderating effect of intrapersonal factors on the association between the built environment and the LTPA of older people in the village of Fuwen, we found that fitness acted as a moderator of the influence of walking facilities on the LTPA of older people. The regression coefficient of walking facilities was significantly negative (Table 6), and the regression coefficient of fitness × walking facilities (Table 7) was also significantly negative. This indicates that the fitness element strengthened the negative association between walking facilities and LTPA. Fitness amplified this influence, which means that older people who had higher fitness requirements and lower perceived walking facilities scores took part in more LTPA. The explanation for this result is similar to that for the negative association of walking facilities with LTPA. Due to the lower SES, walking infrastructure is poor in the village. Those older people who engaged in more LTPA and had higher fitness improvement requirements may have experienced the walking facilities more frequently and might have given a lower perceived score of walking facilities in the interview. In other words, improving fitness was a main motivation for LTPA among older people. Meanwhile, those older people with higher LTPA levels might have a deeper impression of the poor walking facilities. The results of the moderating effect demonstrated that intrapersonal factors can act as moderators of the association of the built environment with the LTPA of older people, and they might either strengthen or weaken the association to some extent.

### 4.5. Strengths and Limitations

The strength of this study lies in its investigation of the effect of built environment and intrapersonal elements on the LTPA level of older people in a Chinese village, and it also demonstrates the moderating effect of intrapersonal factors. There are some limitations to this study. First, the possible influencing factors of the LTPA of older people are numerous, and this study only explored the effect of intrapersonal and built environment factors. Interpersonal factors, like social support, are also important factors of LTPA. For example, Marthammuthu et al. surveyed 1221 community-dwelling older women and suggested that older women with an increase in their social interaction score were more likely to have higher physical activity levels [20]. Second, the significant factors of older people’s LTPA are only effective in the village of Fuwen based on the data collected in this study. Caution should be taken when extending the results to other Chinese villages. Generally, confirmation of the influencing factors of older people’s LTPA should be based on locally collected data. Caution should also be taken when extending the results of this study to other aging populations. Previous studies suggested that the influencing factors of LTPA varied in different age groups [22,52,55]. For example, Zhang et al. suggested that for children, adults, and older people, the physical activity of children and older people was positively associated with pedestrian-friendly features and general safety. Furthermore, the physical activity of adults and older people was positively related to the availability and accessibility of shops/commercial services and parks/open spaces. Lastly, the walkability index was positively associated with physical activity in every age group [22]. The dependent variable was LTPA, and the independent variables were eight elements of the built environment.

Third, the negative association relationship between walking facilities and LTPA was only observed in the participants of this study. Whether this association would change in other villages requires further investigation in future studies. In the regression models, built environment and intrapersonal factors were included. The regression results of this study were useful for investigating the effect of built environment and intrapersonal factors on the LTPA of older people to some extent. However, caution should be exercised when considering whether the regression results would change when other possible influencing factors, such as demographic variables, are included in the models. For future related studies, possible study directions include investigating the influencing factors of older people’s LTPA, such as intrapersonal factors, interpersonal factors, and environmental factors, as well as the interactions among these factors. Comparative studies of the influencing factors of older people’s LTPA between urban and rural areas and across diverse age groups could also be potential research directions.

## 5. Conclusions

The environmental factors of older people’s LTPA in the village of Fuwen included traffic safety, walking facilities, and crime safety. Better traffic and crime safety were associated with higher levels of LTPA among older people. The negative effect of walking facilities on LTPA might be attributable to the fact that older people with higher LTPA levels more frequently experienced poor walking facilities. The intrapersonal factor influencing older people’s LTPA was fitness. Older people who were more eager to improve their fitness engaged in more LTP. Fitness also acted as a moderator of the effect of environmental factors on older people’s LTPA, amplifying the negative association between walking facilities and the LTPA of older people. For local policymakers, improving traffic and crime safety and enhancing the publicity of the health benefits of physical activity would be beneficial in motivating more older people to take part in LTPA. These measures would help maintain the physical and mental health of older people living in the village, and align with the rural revitalization strategy proposed by the Chinese government.

## Figures and Tables

**Table 1 healthcare-13-01244-t001:** Demographic characteristics of participants from Fuwen village (n = 514).

Variable	n	%	Mean ± SD
Age			67.5 ± 21.3
60–69 years	344	66.9	
70–79 years	111	21.6	
≥80 years	59	11.5	
Gender			
Men	260	50.6	
Women	254	49.4	
BMI			23.2 ± 3.1
<18	7	1.4	
18–24	320	62.2	
>24	187	36.4	
Self-reported health status			
Excellent	15	3.0	
Good	396	77.0	
Bad	103	20.0	
Motion sickness			
Yes	154	30	
No	360	70	
Age			67.5 ± 21.3
60–69 years	344	66.9	
70–79 years	111	21.6	
≥80 years	59	11.5	
Gender			
Men	260	50.6	
Women	254	49.4	
Self-reported health status			
Excellent	15	2.9	
Good	396	77	
Bad	103	20	
Lower extremity motor system disease			
Yes	154	30	
No	360	70	

Notes: BMI represents body mass index, and the unit is kg/m^2^; SD indicates standard deviation; 95%CI is the 95% confidence interval value.

**Table 2 healthcare-13-01244-t002:** Description of participants’ leisure-time walking, leisure-time moderate and vigorous physical activity, and LTPA (n = 514).

Variable	Mean ± SD	95%CI
Leisure-time walking	814.0 ± 546.1	0–1980
Leisure-time moderate and vigorous physical activity	533.9 ± 851.9	0–2655
LTPA	1347.9 ± 1132.7	0–4452

Note: The unit of leisure-time walking, leisure-time moderate and vigorous physical activity, and LTPA is MET minutes per week. MET is metabolic equivalent score. SD means standard deviation. 95%CI is the 95% confidence interval value.

**Table 3 healthcare-13-01244-t003:** Description of self-reported scores of intrapersonal factors.

Variable	Mean ± SD	95%CI
Social	2.67 ± 0.52	1.33–3.00
Fitness	2.77 ± 0.45	2.00–3.00
Medical	2.79 ± 0.42	1.67–3.00
Benefits/Challenges	2.70 ± 0.47	1.60–3.00
Recognition	2.71 ± 0.46	1.78–3.00

Note: SD means standard deviation. 95%CI is the 95% confidence interval value.

**Table 4 healthcare-13-01244-t004:** Description of perceived scores of built environment factors.

Variable	Mean ± SD	95%CI
Traffic safety	4.30 ± 1.08	2.0–5.0
Street connectivity	4.27 ± 1.07	2.0–5.0
Walking facilities	4.30 ± 1.03	1.67–5.0
Access to services	4.26 ± 1.11	2.0–5.0
Crime safety	4.58 ± 0.69	3.0–5.0
Aesthetics	4.47 ± 0.80	2.47–5.0
Residential density	56.77 ± 13.04	14.88–65.0
Land use mix diversity	5.45 ± 1.20	1.60–6.20

Note: SD means standard deviation. 95%CI is the 95% confidence interval value.

**Table 5 healthcare-13-01244-t005:** Regression results of intrapersonal factors with LTPA.

Variable	B	SE	Beta	*p*
Social	185.83	156.36	0.29	0.24
Fitness	535.40	212.14	0.85	0.012 *
Medical	−270.65	224.97	−0.43	0.23
Challenges/Benefits	347.03	293.50	0.54	0.24
Recognition	−310.58	270.31	−0.49	0.25

Notes: B represents the regression coefficient, SE indicates the standard error, Beta is the standardized beta coefficient, and * represents a significant association (*p* < 0.05). The F value of this model was 143.82, *p* < 0.001, and the adjusted R^2^ was 0.581.

**Table 6 healthcare-13-01244-t006:** Regression results of built environment factors with LTPA.

Variable	B	SE	Beta	*p*
Traffic safety	381.01	167.34	0.96	0.02 *
Street connectivity	−70.29	158.517	−0.18	0.66
Walking facilities	−524.55	192.07	−1.32	0.007 *
Access to services	263.46	161.74	0.66	0.10
Crime safety	226.08	113.60	0.60	0.047 *
Aesthetics	114.95	161.69	0.30	0.48
Residential density	−2.84	7.06	−0.09	0.69
Land use mix diversity	−53.26	39.29	−0.169	0.18

Notes: B represents the regression coefficient, SE indicates the standard error, Beta is the standardized beta coefficient, and * represents a significant association (*p* < 0.05). The F value of this model was 88.51, *p* < 0.001, and the adjusted R2 was 0.577.

**Table 7 healthcare-13-01244-t007:** Regression results of built environment factors with LTPA with the moderating effect of an intrapersonal factor.

Variable	B	SE	Beta	*p*
Fitness	1261.21	161.72	2.01	<0.001 *
Traffic safety	998.48	849.62	2.52	0.24
Street connectivity	−317.09	792.19	−0.79	0.69
Access to services	−524.67	671.35	−1.31	0.44
Crime safety	541.78	494.93	1.43	0.27
Aesthetics	−293.53	852.73	−0.76	0.73
Residential density	3.63	42.99	0.12	0.93
Land use mix diversity	−241.28	188.34	−0.76	0.73
Fitness × Traffic safety	−306.11	332.82	−2.21	0.34
Fitness × Street connectivity	153.38	303.07	1.10	0.61
Fitness × Walking facilities	−156.81	69.61	−1.13	0.025 *
Fitness × Access to services	323.10	257.79	2.32	0.21
Fitness × Crime safety	−295.41	195.77	−2.22	0.13
Fitness × Aesthetics	55.68	312.70	0.41	0.86
Fitness × Residential density	−1.31	15.42	−0.12	0.93
Fitness × Land use mix diversity	18.12	71.95	0.16	0.80

Notes: As an intrapersonal factor significantly associated with LTPA, fitness was included in the model. B represents the regression coefficient, SE indicates the standard error, Beta is the standardized beta coefficient, and * represents a significant association (*p* < 0.05). The F value of this model was 53.42, *p* < 0.001, and the adjusted R^2^ was 0.62. Walking facilities was excluded from this model by SPSS due to collinearity.

## Data Availability

The original contributions presented in the study are included in the article; further inquiries can be directed to the corresponding author.

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
