# Peer review of "Effect of Environmental and Intrapersonal Factors on Leisure-Time Physical Activity of Chinese Rural Older People in Zhejiang Province"

_healthcare, 2025, doi:10.3390/healthcare13111244_

Round 1
Reviewer 1 Report
Comments and Suggestions for Authors
I would like to appreciate your hard work in carrying out this applied and informative research.
The study subject is crucial given the increasing number of older adults and the significant impact of physical activity on their health and well-being.
Comments
- In the Introduction and Discussion sections, replace secondary references (reviews and systematic reviews) with direct citations from original studies, wherever possible. This will enhance the credibility and clarity of the manuscript, providing readers with a more accurate basis for the claims made.
- In the Methods section, provide further details on regression assumption checks. Also, clearly justify the utilization of multivariate regression models for the analyses, thereby improving the rigor of the study methodology.
- In the Methods section, a significant proportion of the text is dedicated to details of the questionnaires. This content may be presented as a supplementary file, increasing the readability and engagement of your paper.
- In the Discussion section, elaborate on how the findings of this study could be generalized to other aging populations. This will effectively grasp the strengths and limitations of your study regarding external validation.
Please, verify the accuracy of the verb tenses in the various sections of the manuscript.
Author Response
We would like to thank the reviewer 1 for the careful analysis he/she has done of our manuscript and the positive evaluation of this paper again.
Reviewer 1
I would like to appreciate your hard work in carrying out this applied and informative research.
The study subject is crucial given the increasing number of older adults and the significant impact of physical activity on their health and well-being.
Comments
- In the Introduction and Discussion sections, replace secondary references (reviews and systematic reviews) with direct citations from original studies, wherever possible. This will enhance the credibility and clarity of the manuscript, providing readers with a more accurate basis for the claims made.
Response 1: thank you for the suggestion. We have added direct citations wherever possible. For this purpose, we added the new references 28, 29, 49, 50, 53, 56. Please see lines 75-78, 312-314, 342-344, 371-373.
- In the Methods section, provide further details on regression assumption checks. Also, clearly justify the utilization of multivariate regression models for the analyses, thereby improving the rigor of the study methodology.
Response 2: We added more details on the regression method used in this study. Please see lines 225-231.
- In the Methods section, a significant proportion of the text is dedicated to details of the questionnaires. This content may be presented as a supplementary file, increasing the readability and engagement of your paper.
Response 3: Thank you for the suggestion. We have moved the details of questionnaires into the appendix section.
- In the Discussion section, elaborate on how the findings of this study could be generalized to other aging populations. This will effectively grasp the strengths and limitations of your study regarding external validation.
Response 4: according to your suggestion, we have added this limitation. Please see lines 412-414.
Reviewer 2 Report
Comments and Suggestions for Authors
Thank you for submitting to Healthcare. The authors have done an interesting study using an interesting topic. However, I would like to request further revision.
Abstract
Methods: Reflecting the title, the authors should briefly describe what variables were measured for environmental and intrapersonal factors. The authors should also provide more details on the research methods to provide better understanding to the readers.
Results: Providing p values ​​would be a way to increase the reliability of this article.
Introduction: Overall, it is well written, but needs to be organized a bit more.
The introduction is ultimately necessary to explain the purpose of the study. Healthcare does not have guidelines for the length of the introduction. However, it is right to cite previous studies to support the necessity of this study, but the current introduction is too long. It needs to be reduced by about 20%. In particular, it describes too much about the increase in the elderly population until physical activity is mentioned (line 41). The author wrote that the seriousness of the increase in the elderly population has already been emphasized by many literatures, so most readers already know it.
And divide the paragraph at line 41.
Up to line 61, the author repeats general information about the elderly - physical activity - general information about the elderly - physical activity. It is necessary to reduce the volume and organize it into more systematic paragraphs.
After line 62, the content about built environmental and intrapersonal factors is well written.
It is recommended that the purpose of the study be written more specifically and a hypothesis be added.
Research method
“2.2 The questionaries used in data collection” is too much. It is recommended that the section be divided into 2-3 and divided into the areas of PA, built environmental, and intrapersonal factors.
Analysis
I hope that there will be more to the current analysis.
1. Conduct multi-regression, but select stepwise.
2. The variables selected through multi-regression (built environmental and intrapersonal factors) should be used to compare physical activity according to their status. Through this process, readers can see how much difference there is in physical activity participation according to environmental and intrapersonal high and low.
The main message of this study is to find factors related to built environmental and intrapersonal factors to ultimately increase physical activity, and to solve them to increase PA.
Discussion
The study was limited to Zhejiang Province. If not Chinese readers, people from other countries do not know much about Zhejiang (e.g., cities or provincial small cities/industrial cities or rural or fishing villages). Therefore, the author should describe the regional characteristics of Zhejiang by connecting them. Then, the author should add more writing about the characteristics, culture, and social system of Zhejiang.
Author Response
We would like to thank the reviewer 2 for the careful analysis he/she has done of our manuscript and the positive evaluation of this paper again.
Reviewer 2
Thank you for submitting to Healthcare. The authors have done an interesting study using an interesting topic. However, I would like to request further revision.
Abstract
Methods: Reflecting the title, the authors should briefly describe what variables were measured for environmental and intrapersonal factors. The authors should also provide more details on the research methods to provide better understanding to the readers.
Results: Providing p values ​​would be a way to increase the reliability of this article.
Response 1: according to your suggestion, we have modified the method and results parts of introduction. Please see lines 17-22.
Introduction: Overall, it is well written, but needs to be organized a bit more.
The introduction is ultimately necessary to explain the purpose of the study. Healthcare does not have guidelines for the length of the introduction. However, it is right to cite previous studies to support the necessity of this study, but the current introduction is too long. It needs to be reduced by about 20%. In particular, it describes too much about the increase in the elderly population until physical activity is mentioned (line 41). The author wrote that the seriousness of the increase in the elderly population has already been emphasized by many literatures, so most readers already know it.
And divide the paragraph at line 41.
Up to line 61, the author repeats general information about the elderly - physical activity - general information about the elderly - physical activity. It is necessary to reduce the volume and organize it into more systematic paragraphs.
Response 2: according to your suggestion, we have shortened the first and second paragraphs. Please see lines 33-64.
After line 62, the content about built environmental and intrapersonal factors is well written.
It is recommended that the purpose of the study be written more specifically and a hypothesis be added.
Response 3: we have added the hypothesis of this paper. Please see lines 111-114.
Research method
“2.2 The questionaries used in data collection” is too much. It is recommended that the section be divided into 2-3 and divided into the areas of PA, built environmental, and intrapersonal factors.
Response 4: we have moved the questionnaires used in this study to the appendix section at the end of the manuscript.
Analysis
I hope that there will be more to the current analysis.
1. Conduct multi-regression, but select stepwise.
2. The variables selected through multi-regression (built environmental and intrapersonal factors) should be used to compare physical activity according to their status. Through this process, readers can see how much difference there is in physical activity participation according to environmental and intrapersonal high and low.
The main message of this study is to find factors related to built environmental and intrapersonal factors to ultimately increase physical activity, and to solve them to increase PA.
Response 5: Thank you for the suggestions. We have added more details on the regression analysis used in this study. Please see lines 225-231. The regression method we conducted was enter based on previous studies. Generally, one regression method was used in one paper, and we might try stepwise in future studies. In the results of regression analysis, the regression coefficient B could indicate the association direction between LTPA and variables.
Discussion
The study was limited to Zhejiang Province. If not Chinese readers, people from other countries do not know much about Zhejiang (e.g., cities or provincial small cities/industrial cities or rural or fishing villages). Therefore, the author should describe the regional characteristics of Zhejiang by connecting them. Then, the author should add more writing about the characteristics, culture, and social system of Zhejiang.
Response 6: Thank you for the suggestions. We have added some introductions of Zhejiang in the method section. Please see lines 121-124.
Round 2
Reviewer 1 Report
Comments and Suggestions for Authors
Thank you for revising the manuscript.
For better clarity, please consider providing additional information regarding comments 2 and 4.
Comments on the Quality of English LanguagePlease review the text for verb tenses, and ensure the symbols in the Tables are formatted correctly.
Author Response
We would like to thank the reviewer 1 for the careful analysis he/she has done of our manuscript and the positive evaluation of this paper again.
1. For better clarity, please consider providing additional information regarding comments 2 and 4.
Response1:
Thank you for the careful suggestions. According to your suggestions, we have modified the manuscript to a larger extent. Absolutely, those changes improve the manuscript quality.
About the comment 4, our opinion is extending the results of this study to other aging populations should also be caution because previous studies have suggested the influencing factors of LTPA varied in age groups. In the limitations, we have added the finding of a previous study to testify it as follows: “For example, Zhang et al suggested that for children, adults, and older people, physical activity of children and older people was positively associated with pedestrian -friendly features and general safety. Furthermore, physical activity of adults and older people was positively related to the availability and accessibility of shops/commercial services and parks/open spaces. Lastly, the walkability index was positively associated with physical activity in every age group” please see lines 412-418.
About the comment 2, we added the description of regression assumption checks as follows: “Before conducting the regression analysis, we operated correlation analysis and confirmed that eight elements of built environment and five elements of intrapersonal factors were significantly related to LTPA level of older people.”
We also modified the description of regression models to make it more clearly as follows: “In the regression model of LTPA with built environment, the dependent variable was LTPA, and the independent variables were eight elements of built environment. In the regression model of LTPA with intrapersonal factors, the dependent variable was LTPA, and the independent variables were five elements of intrapersonal. Multivariate linear regression analysis was also operated to identify the possible moderating effect of intrapersonal factors on the association between LTPA and built environment. In this regression model, the dependent variable was LTPA, and the independent variables were eight elements of built environment and the intrapersonal factors with significantly related to LTPA.” Please see lines 225-235
Furthermore, we noticed the regression models used in this study were limited, so we added some limitations as follows: “In the regression models, built environments and intrapersonal factors were included. The regression results of this study were useful to investigate the effect of built environments and intrapersonal factors on LTPA of older people to some extent. However, Caution should be exercised when considering whether the regression results would change when other possible influencing factors, such as demographic variables, are included in the models.” Please see lines 427-432
2. Please review the text for verb tenses, and ensure the symbols in the Tables are formatted correctly.
Response2:
we have modified the formation of symbols in the tables 1-7 according to the temple of healthcare. Thank you for the careful suggestion.
We have reviewed the text for verb tenses, the changes are as follows:
Line 46, 54, 92, 116, 133, 190, 314, 317, 318 350, 357, 379
We will appreciate it if you could point out other text needed to be modified the verb tenses. Thank you for the valuable review.
Reviewer 2 Report
Comments and Suggestions for Authors
I do not have any comment.
Author Response
Dear reviewer 2,
Thank you for the valuable suggestions.